# A 0.617–2.7 GHz Highly Linear High-Power Dual Port 15 Throws Antenna Switch Module (DP15T-ASM) with Branched-Antenna Technique and Termination Mode

**DOI:** 10.3390/s22062276

**Published:** 2022-03-15

**Authors:** Reza E. Rad, Kyung-Duk Choi, Sung-Jin Kim, Young-Gun Pu, Yeon-Jae Jung, Hyung-Ki Huh, Joon-Mo Yoo, Seok-Kee Kim, Kang-Yoon Lee

**Affiliations:** 1Department of Electrical and Computer Engineering, Sungkyunkwan University, Suwon 16419, Korea; reza@skku.edu (R.E.R.); glylop@skku.edu (K.-D.C.); sun107ksj@skku.edu (S.-J.K.); hara1015@skku.edu (Y.-G.P.); yjjung@skaichips.co.kr (Y.-J.J.); gray@skaichips.co.kr (H.-K.H.); jmyoo@skaichips.co.kr (J.-M.Y.); skkim@skaichips.co.kr (S.-K.K.); 2SKAIChips Co., Ltd., Suwon 16419, Korea

**Keywords:** antenna switch module, DP15T, termination mode, branched antenna, high power

## Abstract

This paper presents a Dual-Port-15-Throw (DP15T) antenna switch module (ASM) Radio Frequency (RF) switch implemented by a branched antenna technique which has a high linearity for wireless communications and various frequency bands, including a low- frequency band of 617–960 MHz, a mid-frequency band of 1.4–2.2 GHz, and a high-frequency band of 2.3–2.7 GHz. To obtain an acceptable Insertion Loss (IL) and provide a consistent input for each throw, a branched antenna technique is proposed that distributes a unified magnetic field at the inputs of the throws. The other role of the proposed antenna is to increase the inductance effects for the closer ports to the antenna pad in order to decrease IL at higher frequencies. The module is enhanced by two termination modes for each antenna path to terminate the antenna when the switch is not operating. The module is fabricated in the silicon-on-insulator CMOS process. The measurement results show a maximum IMD2 and IMD3 of −100 dBm, while for the second and third harmonics the maximum value is −89 dBc. The module operates with a maximum power handling of 35 dBm. Experimental results show a maximum IL of 0.34 and 0.92 dB and a minimum isolation of 49 dB and 35.5 dB at 0.617 GHz and 2.7 GHz frequencies, respectively. The module is implemented in a compact way to occupy an area of 0.74 mm^2^. The termination modes show a second harmonic of 75 dBc, which is desirable.

## 1. Introduction

In recent years, antenna switch modules (ASMs) have seen increasingly extensive use in RF front-end modules, which have a vast range of applications in mobile terminals. This is due to the rapid growth of wireless communication standards at various frequency bands across various mobile systems. Due to the extensive demand for low supply voltage systems as well as the integration of CMOS logic control circuits into chips, silicon-on-insulator (SOI) has become the dominant process used for RF switch applications [1,2,3]. Furthermore, the GaAs pHEMT process is another process which is attractive for this application [4,5,6]. Therefore, the SOI process fits most of the fabrication aid designs.

For partially depleted (PD) SOI process technology, not all of the buried oxide (BOX) layer is doped [7,8]. Therefore, a natural body region is created below the channel. Moreover, if the gate of the field-effect transistors (FETs) is connected to the body or is floating, FETs are categorized as floating body (FB) [9,10] or body-connected (BC) FETs [11,12]. The body of an FB FET is floating but the body of a BC FET is accessible through body contact. There are several works that employ FB devices [13,14,15], and many RF switches are made using BC devices [16,17,18]. In [19], an SP8T is designed based on both FB and BC FETs. A comparison between FB and BC FETs shows that by using FB FETs, a kink effect occurs because of the presence of high voltages. Therefore, electrons drop into the body and the body voltage varies, which degrades linearity. Furthermore, negative biasing is used for both gate and body biasing [19,20,21,22,23]. By using negative biasing, the device junctions remain in reverse bias, which results in lower leakage and higher linearity.

Several studies have aimed to improve performance and reduce area occupation. In [18], a Radio Frequency (RF) switch was introduced with extremely low power consumption by removing the negative charge pump that is essential in similar works. The benefit of the work is a lower area occupation while, due to operating without a negative charge pump, spurious emissions from the oscillator used in the charge pump are surpassed. This is obtained by an intermediate node biasing technique which maintains the switch junctions in reverse bias in order to reduce leakage and improve linearity. However, isolation is reduced due to the intermediate node biasing. On the other hand, in SOI processes, two general devices are used to implement the RF switches.

Therefore, the voltage of the body varies and results in a lower linearity performance. Due to the high data rates in mobile network systems and their wide variety of applications, several RF switches have been proposed with multi-ports and multi-throws. In this study, the design of a Dual Port 15 Throws (DP15T) module that is enhanced with two termination paths for the termination mode is proposed. A branched antenna path is proposed to overcome IL issues arising from the design complexity and the number of throws and ports, while improving the linearity and providing the required area to improve power handling.

## 2. The Proposed DP15T ASM

Figure 1 shows the top block diagram of the proposed DP15T in an RF front-end application. It is shown that two antennas correspond to low bands and mid–high bands as the input ports of the module. The proposed module switches the antennas between 15 throws, which are fed to another front-end module (Low Noise Amplifiers). One antenna is dedicated to low-frequency bands, while the other is dedicated to mid–high-frequency bands. The low-frequency antenna port is connected to seven throws along its antenna path, forming a Single Port 7 Throws (SP7T). The mid–high-frequency antenna port is connected to eight throws through its antenna path, which shapes an SP8T.

### 2.1. The Proposed Building Block

The proposed wide-band DP15T is shown in Figure 2. The proposed RF-switch covers all of the low, middle and high-frequency bands of the Long-Term Evolution (LTE) mobile system by implementing two separated paths for low-frequency bands through the SP7T and middle/high-frequency bands through the SP8T. This feature facilitates the implementation of an RF front-end for LTE and similar mobile systems applications. Furthermore, the proposed structure is enhanced by two termination paths for each antenna path. When the antennas of the DP15T are close enough to another antenna of an adjacent device transmitting a high power RF signal, the possibility of damage is very high for the DP15T while it is in sleep mode and all of the throws are off. Therefore, the proposed structure is safer than similar works without a termination mode. To realize the proposed block-diagram in real-world applications, a long antenna is required due to the complexity of the work and the number of throws, which with the conventional straight antennas in other works in the literature results in very high loss and lower linearity. The other contribution of this paper is a proposed branched antenna for such a complex structure to provide a low Insertion Loss (IL) and high linearity for the proposed DP15T.

### 2.2. The Proposed Termination Circuit

The proposed termination circuit is shown in Figure 3. The circuit of the termination mode is designed to suppress any unwanted high-power RF signal transmitted by any nearby antenna of any other devices which might damage the RF switches. One of the most critical concerns in the design of every RF switch is the power-handling capability of the RF switch. For the termination-mode’s circuit, a relatively high power handling is required.

Therefore, the maximum power handling must be considered as follows [19]:(1)$$P_{\max}\;=\;2\;\frac{V_{\max}^{2}}{2Z_{0}}\;=\;\frac{n^{2}\;[\min(V_{\mathrm{bdds}},V_{\mathrm{DS},\max}){]}^{2}}{2Z_{0}}$$
where n is the number of stacked-FETs and V_bdds_ is the breakdown voltage of drain-to-source. The power handling which is concerned with the off branches is directly proportional to the number of stacks of the shunt and series stacks. The proposed termination circuit is designed with 10 stacks in series. This is due to the relevant power level of the RF signal at 26 dBm and 902.4 MHz. In this case, the second harmonic is desired to be attenuated to 70 dBc in comparison with the main tone. This is obtained by a width of 1.8 mm and a 200 kΩ gate and body resistance for each switch to minimize the gate and body leakages, respectively. To maximize gate isolation, which has a dominant effect on linearity and lower harmonics, an additional series resistor is placed with the shared node of the gate resistances and the bias voltage of the gate. A 50 Ω load is placed after the last stack on the ground. This will provide an absolute conjugate matching and terminate the received high-power RF signal to the ground during termination mode. By receiving such a high-power RF signal, a high ac current will path through the resistor. Therefore, the breakdown of the resistor is related to implementing the resistor with several parallel segments in the layout in order to divide the current through the parallel segments and prevent its breakdown. It is possible that both the antennas receive such high power when both the SP7T and SP8T are in termination mode. Therefore, to avoid overcurrent at the ground pad, a separate ground pad is used for each termination rout, which maximizes the safety of the operation during termination mode.

To guarantee that the junctions of the transistors are reverse-biased when they are off, which increases the overall linearity of the work by avoiding any leakage current through the forward biasing of the junctions, negative biasing is used. While the switches are on, their gates are biased with 1.8 V and their bodies are biased with 0 V, but for off-operation both the gates and bodies are biased with −1.8 V.

### 2.3. The Proposed Branched Antenna

Due to the numerous throws (15 throws) and two termination paths, which produce enormous parasitics at high frequencies, the design of the antenna is critical. The other aspect of the antenna is its length, which can be as long as 1.5 mm for the SP8T path. This length of antenna, in addition to the high parasitic components in the higher frequencies, introduces a high loss for the higher frequencies. This is more challenging when the compactness of the work is restricted to a small area of occupation of the die area. In this section, the implementation of the antenna is analyzed in terms of its length and electromagnetic field distribution and its effect on the inductance at the input of each throw, which must have consistent behavior in order to obtain a consistent operation performance among the outputs of the throws.

One of the most challenging aspects of the proposed design is the length of the antenna. The length and width of the antenna require accurate analysis with an Electro-Magnetic (EM) modeling tool. Because the metal layers in the SOI-CMOS process are flat, the inductance (L) of the antenna line follows as below in the μH dimension:(2)$$L\;=\;2\;\times \;10^{-4}\;\times \;l\;\times \;\ln\left\{\frac{2l}{(w\;\times \;10^{-4})\;+\;t}\right\}\;+\;0.5\;+\;0.2235\left\{\frac{(w\;\times \;10^{-4})\;+\;t}{l}\right\}$$
where l is the length of the antenna path in mm, w is the width of the metal layer in micro-meters, and t is the thickness of the metal layer in mm. In the SOI-CMOS process, metal thickness is in micro-meters. Therefore, the inductance of the antenna line varies significantly for different throws with different lengths (l). Equation (2) shows that the length of the antenna has a dominant effect on the inductance effect. This conclusion is used to place the input of the throws further from the antenna pad (port) in comparison with the conventional antennas that are connected to the throws as closely as possible. The reason for the importance of the inductance effect for the proposed DP15T is the presence of high capacitive parasitic components at the high frequencies. By introducing a higher series inductance before the input of each throw, a better conjugate matching is provided for each throw.

Further analysis was performed on the conventional antenna method to achieve the proposed branched-antenna implementation. Figure 4 shows the electromagnetic analysis over the mid–high-band antenna path (SP8T) in a conventional implementation, which was obtained using the RF-momentum tool of an ADS simulator. The magnetic field distribution, which was directly proportional to the linear current density with a dimension of Ampere/meter (A/m), was simulated over the antenna path. It is also expressed as ampere-turn per meter to reflect the magnetic field strength. Therefore, with the conventional direct implementation, the magnetic field varies significantly from the first antenna connection (throw 1) to the last antenna connection (throw 8). In terms of the effect of the magnetic field strength, the relationship between magnetic flux (B) and H is:(3)B = μ.H
where µ is the permeability of the antenna path’s metal. Based on Faraday’s law, the loop of the antenna path forms an area (A) and the loop defines a magnetic flux. Furthermore, the relationship between magnetic flux and inductance is:(4)B = L.i
where i is the current in the antenna path. Equations (3) and (4) result in:(5)$$L\;=\;\frac{\mu .H}{i}$$

Equation (5) reflects that, due to the ununified electromagnetic field distribution at the antenna connection from throw 1 to throw 8, the inductance at these connection nodes is also inconsistent, which will result in an inconsistency in the operation performance between the throws.

Figure 5 shows the proposed branched antenna implementation for the mid–high-band path (SP8T). Similarly, the electromagnetic simulation is performed over the antenna path. It is shown that the electromagnetic field distribution has a consistent behavior at the antenna connection for each throw. In fact, the branches which are connected to the main antenna path have the same similar behavior and inductance as a consequence.

The other aspect of this particular implementation is that it allows for a more compact layout implementation, as the top layout shows in Figure 6. Meanwhile, it keeps the value of IL lower than the design with the conventional antenna rout. Moreover, due to the placement of the first throws at a longer distance from the antenna port, the inductance effect of the antenna is improved for these ports and causes a lower IL for these throws.

### 2.4. Circuit Implementation

To implement the DP15T, 15 RF switch arms are required to realize the IL, isolation, power handling, and linearity requirements. The proposed RF switch arm is shown in Figure 7. When one of the throws of SP7T or SP8T is connected to an antenna, the series stacks are ON, the gate bias voltages are 1.8 V, and the body bias voltages are 0 V; while the shunt stacks are off, their gate bias is −1.8 V and their body is biased with −1.8 V as well. For the other throws which are disconnected, their switch arms are OFF, with a −1.8 V gate and body biasing for the series stacks, and 1.8 V and 0 V biasing for the gate and body of their shunt stacks.

To provide a low IL, the width of the transistors, the gate and body resistors, and the level of the bias voltage are the dominant parameters in the design of the switch arm. Meanwhile, due to the compactness of the design, the area occupation is a limiting parameter. Therefore, a 2.3 mm width and 200 kΩ gate and body resistors formed a switch cell. The role of the drain–source resistor is to keep drain–source voltages close to each other. Through a further increase in the width of the transistors, parasitic capacitance increases, and this will affect both the power handling and isolation parameters. A good isolation means that, when an antenna’s RF signal is connected to a throw, other throws be well isolated. By having a larger width, the parasitics of the series stacks increase, which results in greater signal coupling to the throws which are in the off-state.

To enhance the linearity performance, large gate and body resistors are used for the switches (RG_SE and RG_SH for the series and shunt stacks, respectively), while the maximum value of the gate resistors is restricted by the switching time (the required time for the module to turn off a throw and turn on another throw). In addition to the gate resistor, a series resistor with the gates (RGCM) is used, which increases the effectiveness of the stated trend. Additionally, negative biasing is used for both the gate and body biasing [19,20,21,22,23].

For the proposed throw, 9 series and 9 shunt stacks were designed, with a 2.3 mm width for each transistor of the series and 0.7 mm for the shunt stacks. Nine series stacks with a 2.3 mm width are required for both intermodulation (IMD) improvements as well as for maximum power handling. Increasing the size of the series stacks introduces a larger parasitic capacitance, which influences the isolation of the switch when the throw is off. The same is true for the shunt stacks in terms of the number of stacks and power handling. The width of the shunt stacks has a direct influence on the linearity. With higher width for the shunt stacks, the shunt parasitic capacitance at the output node will be larger, and the linearity will degrade.

Even though the forward biasing of the device junction diodes is prevented by using negative biasing, spurious emissions caused by the oscillator which is used in providing negative biasing are a potential source of linearity degradation. Therefore, in the design of the floor-plan of the top layout (Figure 6), the oscillator must be placed far away from the RF signal lines. Furthermore, the overlap of the bias and control routings must be shielded from the RF metal lines with a shielding technique. This also prevents any linearity degradation due to the coupling effect when RF signals are large. The final stage of the top layout implementation shown is to model of all the antenna lines with an EMX tool to provide the most accurate results. Equation (1) shows that, to enhance the level of maximum power, the number of the stacks must be increased. By increasing the number of stacks, the large amplitude signal will be divided over a larger number of stacks, which results in a lower voltage tension over the device junctions. The power handling of the proposed DP15T is analyzed by increasing input power through the antenna and observing the harmonic, as shown in Figure 8. At the edge of maximum power (37 dBm), the devices will break and the harmonics will suddenly increase.

## 3. Measurement Results

In order to measure IMD2, two blocker tones were used, the first blocker at 1.95 GHz and 836.5 MHz, with 24 dBm and 10 dBm power, and the second blocker at 4.09 GHz, 190 MHz, 1718 MHz, and 45 MHz, with a −15 dBm power. Similarly, for IMD3, the first blocker was used at 1.95 GHz and 836.5 MHz, with 24 dBm and −15 dBm, and the second blocker at 6.04 GHz, 1.76 GHz, 2554.5 MHz, and 791.5 MHz, with a −15 dBm power.

The experimental results show a maximum of −100 dBm for both the IMD2 and the IMD3. The second and third harmonic are also measured using a 25 dBm tone in every band. Figure 9 and Figure 10 show the measured SP parameters of the module, which were used to obtain the IL and isolation for the LB and MHB paths, respectively. The data are plotted from the extracted SP file using a network analyzer. For the low-frequency bands (SP7T path) at 617 MHz and 960 MHz, the obtained IL is 0.35 dB and 0.39 dB, respectively, and the obtained isolation is 49 dB and 45 dB, respectively. For the middle- and high-frequency bands (SP8T path), at 1400 MHz and 2700 MHz, the obtained IL is 0.53 dB and 0.92 dB, respectively, and the obtained isolation is 40.8 dB and 35.5 dB, respectively. Therefore, for the higher frequencies, both performance parameters degrade, which is due to the stronger parasitic components become larger with increased frequency. Due to this complication, the design of the antenna and its analysis discussed here have a critical role in meeting the obtained measurement results for the IL and the isolation of the module.

The experiment performance parameters of the module are listed in Table 1. A comparison between the design and similar works was conducted, which shows a higher linearity in comparison with other works in terms of the 2nd and 3rd harmonics, which is due to the series resistor with the gate resistors in the throws; the branched antenna design, which unified the magnetic field at the input of the throws; and the overall optimized design. The other obvious advantage of this work compared with similar works is the compactness of the work in the presence of several RF connections. The area occupation of the work is completely reduced due to the branched antenna and the new floor plan in the top layout. A Figure Of Merit (F.O.M.) is introduced:(6)$$F.O.M.\;=\;\frac{A}{N_{\mathrm{Throws}}\;+\;N_{\mathrm{Ports}}}$$
where A is the area occupied by the die of the chip, and N_Throws_ and N_Ports_ are the number of throws and ports, respectively. The results are obtained while the supply voltage of the module is 1.8 V and the other works gain an advantage from a 2.5 V supply voltage, which has a direct relationship with a better linearity and SP-parameter results. The other advantage of the proposed module is that it has a termination mode, in contrast to other works.

## 4. Conclusions

In this paper, a DP15T antenna switch module is proposed. A branched antenna is enhanced to improve the IL and linearity. Two termination paths are implemented for safety aids when the module is not operating. The measurement results show a maximum IMD2 and IMD3 of −100 dBm. IL is kept at a low value considering the complexity of the design due to the proposed antenna path, for which electromagnetic analysis and inductance calculations were used in order to yield a precise design. Eventually, a power handling of 35 dBm was reached.

## Figures and Tables

**Figure 1 sensors-22-02276-f001:**
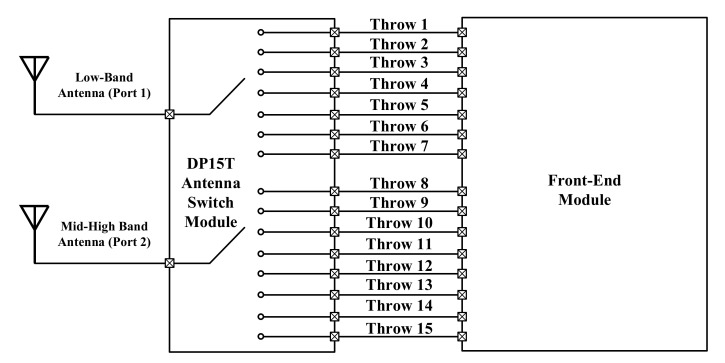
The proposed DP15T antenna switch module in a RF front-end application for mobile communication systems.

**Figure 2 sensors-22-02276-f002:**
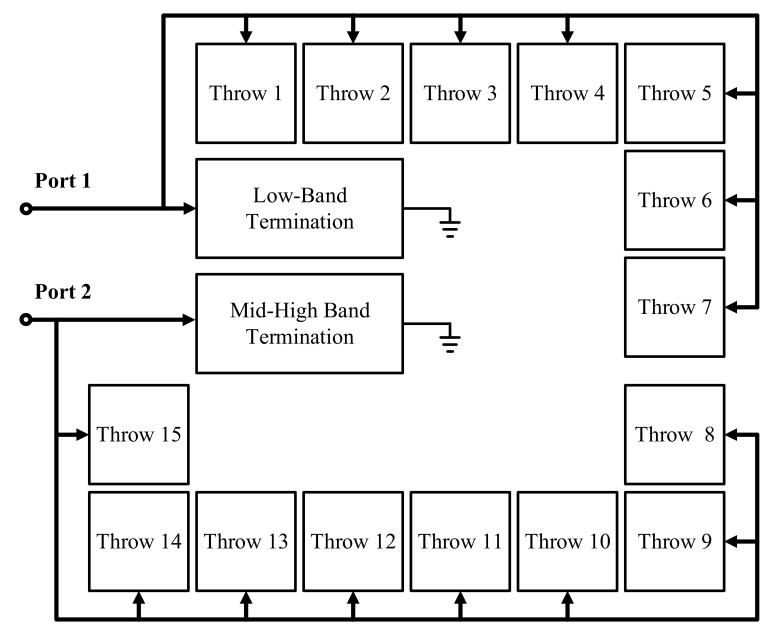
The top RF building block of the proposed DP15T ASM with termination paths.

**Figure 3 sensors-22-02276-f003:**
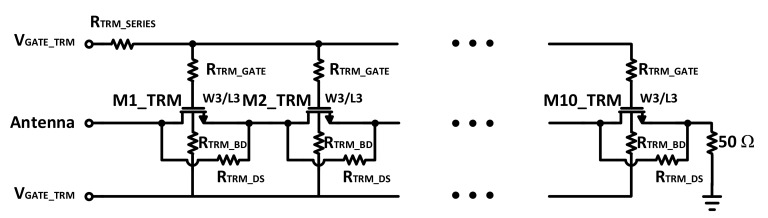
The proposed termination path for the termination mode.

**Figure 4 sensors-22-02276-f004:**
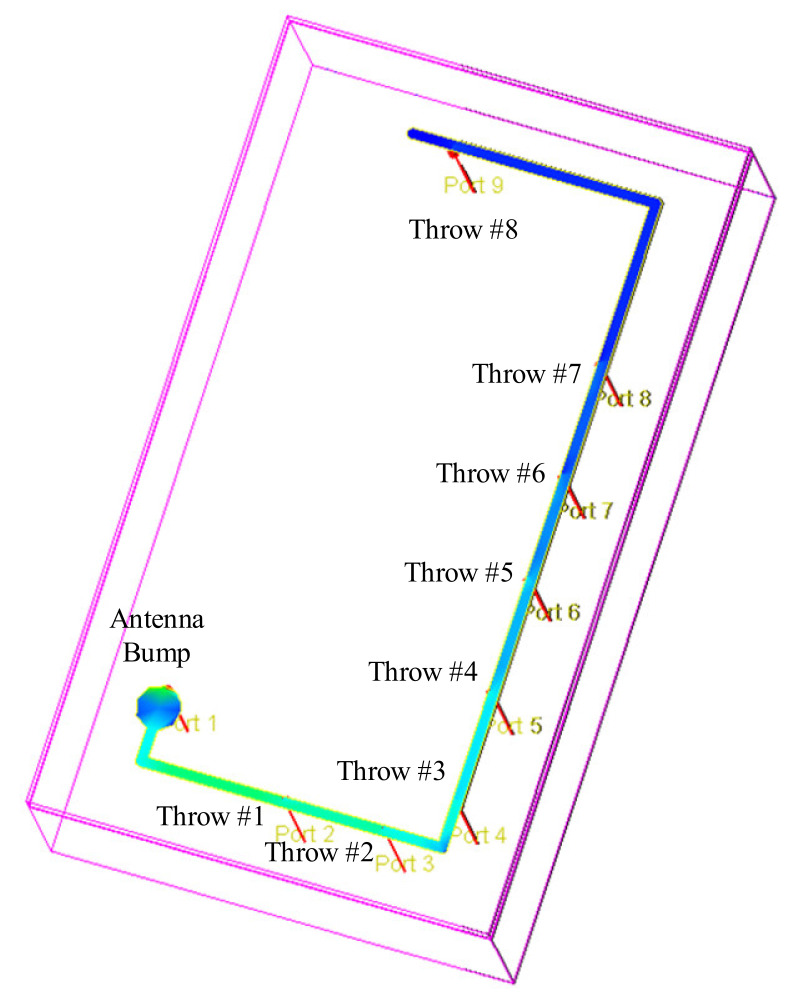
Electromagnetic analysis over the conventional direct antenna connection.

**Figure 5 sensors-22-02276-f005:**
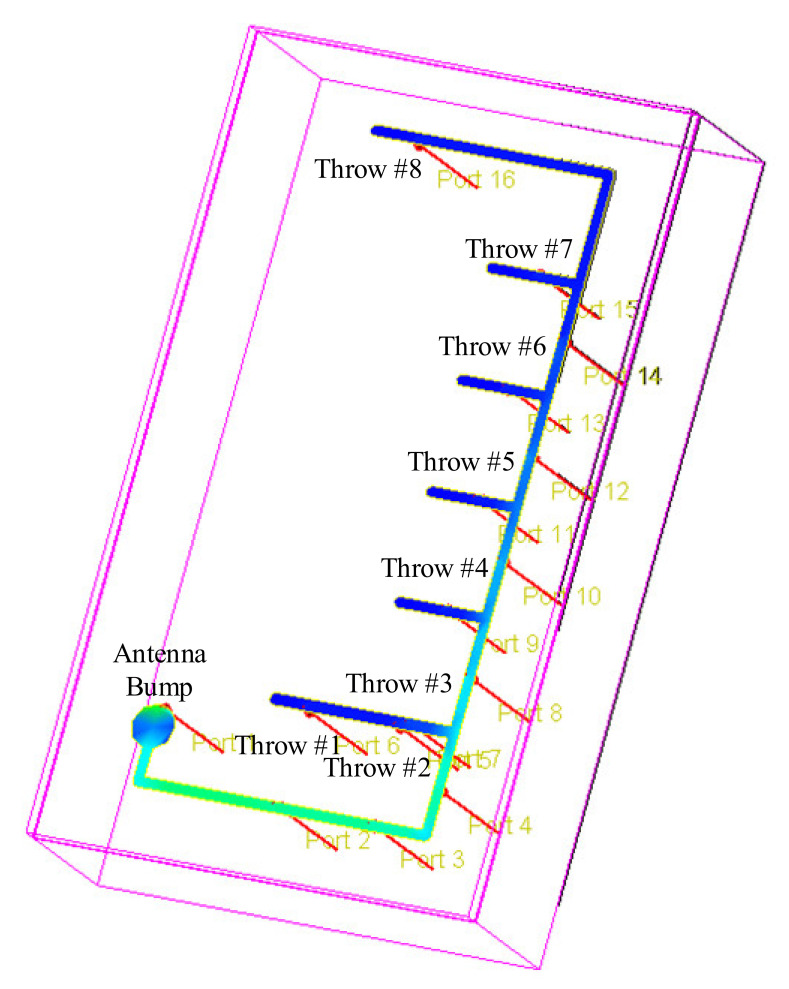
Electromagnetic analysis over the proposed branched antenna paths.

**Figure 6 sensors-22-02276-f006:**
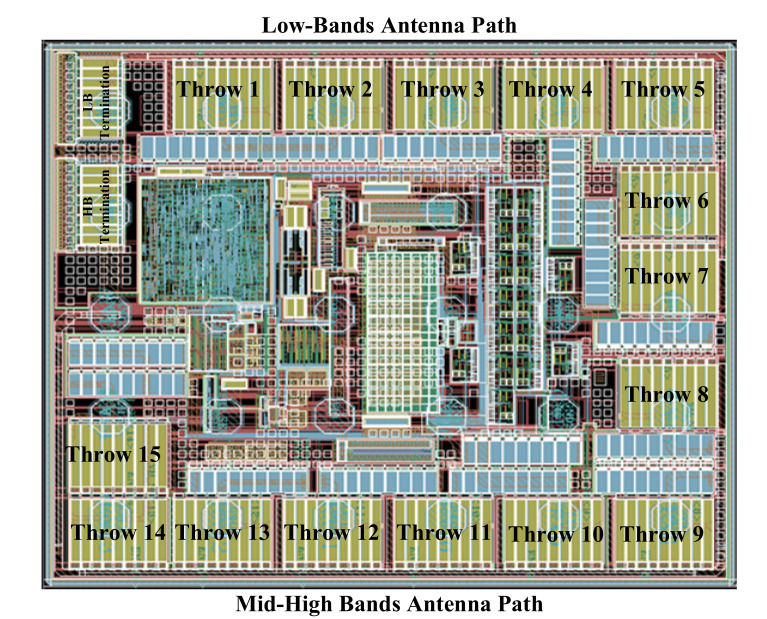
The top layout of the proposed module, illustrating the SP7T and SP8T RF switches and their corresponding terminations paths, TRM1 and TRM2, respectively.

**Figure 7 sensors-22-02276-f007:**
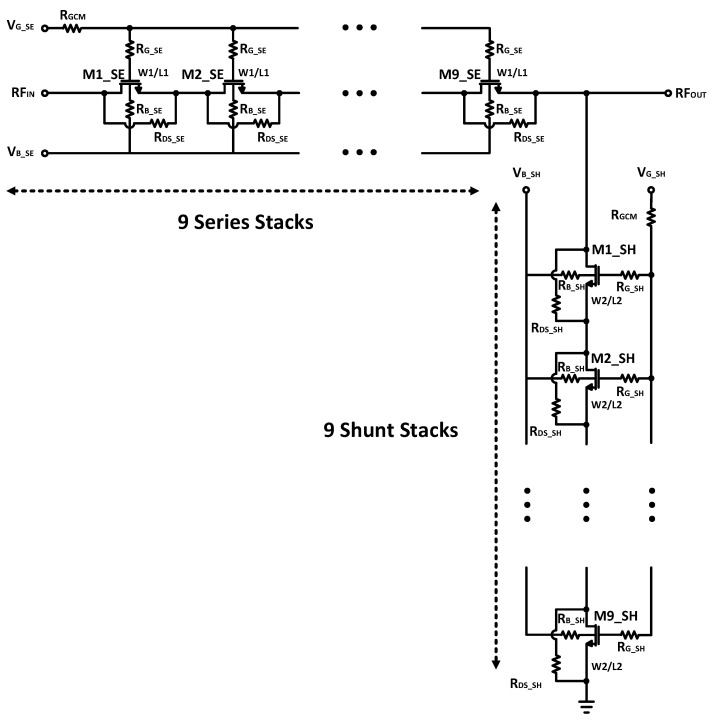
The proposed RF switch arm with 9 series stacks and 9 shunt stacks with a linearity enhancement design.

**Figure 8 sensors-22-02276-f008:**
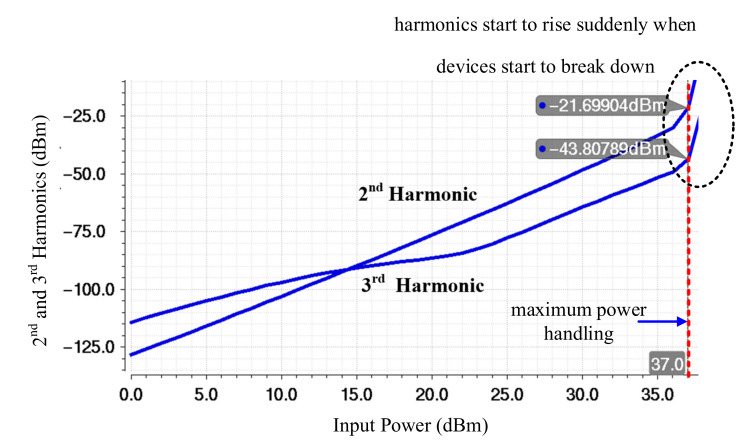
Analysis of power handling achieved by sweeping input power and monitoring the harmonics.

**Figure 9 sensors-22-02276-f009:**
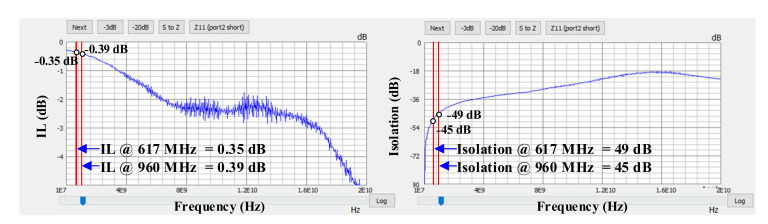
Measurement results of the S-Parameters for the low bands from 617 MHz to 960 MHz.

**Figure 10 sensors-22-02276-f010:**
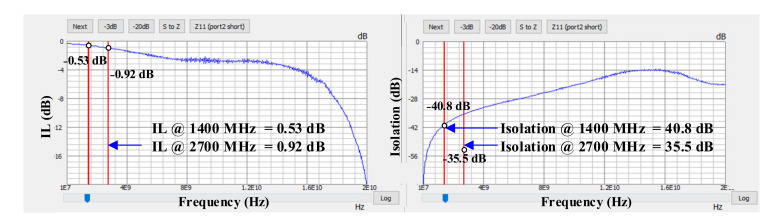
Measurement results of the S-Parameters for the mid and high bands from 1.4 GHz to 2.7 GHz.

**Table 1 sensors-22-02276-t001:** Comparison table for various TIA structures.

Parameters	[19]	[22]	[23]	This Work
Supply Voltage (V)	2.5	2.5	2.5	1.8
Frequency Range (GHz)	0.9–1.9	0.9–1.9	0.9–1.9	0.617–2.7
Structure	SP8T	SP9T	SP8T	DP15T (SP8T + SP7T)
Insertion Loss (dB)	0.53–0.65	0.42–0.55	0.65–0.78	0.34–0.92
Minimum Isolation (dB)	38.4–29.3	26–20	45–37	49–35.5
2nd Harmonic (dBc)	−78–−75.5	−90–−90	−80–N/A	−92–−89
3rd Harmonic (dBc)	−81.6–−79.9	−76–−82	−85–N/A	−97.2–−88
Power Handling (dBm)	38.4	36	35	35
Process Technology	SOI-CMOS	SOI-CMOS	SOI-CMOS	SOI-CMOS
Maximum 2nd and 3rd IMDs (dBm)	N/A	N/A	N/A	−100
Size (mm^2^)	1.21	1.52	N/A	0.74
Number of total RF connections (ports and throws)	8	6.2	20.7	2.71
Termination Mode	NO	NO	NO	YES
F.O.M.	0.134	0.152	N/A	0.043

## Data Availability

Not applicable.

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
