# Peer review of "A 0.617–2.7 GHz Highly Linear High-Power Dual Port 15 Throws Antenna Switch Module (DP15T-ASM) with Branched-Antenna Technique and Termination Mode"

_sensors, 2022, doi:10.3390/s22062276_

Round 1

Reviewer 1 Report

The authors have to clarify where the novelty is? The work seems to be a very good engineering work, designed step by step and there are measurements results too, which is very positive.  But a real scientific work is missing or hidden. Like  for example any optimazation details. Or an extension to a menthod. Any new design is potentially a new paper. Why this one? And how the authors were guided to the spesific configuration? These questions must be clarified. The whole analysis is rather a description af various steps by adding components meaning a study than a scientific work.

What's the difference of the paper from a study? If it is a study it must be written at title. If not the scientific steps ahead must given.

How the authors was driven to the specific configuration? The novelty is at the design, at construction at the proposed termination mode or there is something else? Obviously any difference at design from an existing one is a potential  new paper.

Details must be given to any optimization procedure used.   At page 3 is written that "the reason of these sizings is the tight trade off among other performance parameters....." A more detail explanation needed to avoid my general comment that the paper is description of steps. Besides  the term "sizing" sound not very scientifically.

Fig. 4 do not show any analysis. Its a figure of  field distribution. Analysis is missing at the paper.

Author Response

The reviewer’s concerns are valid and we have the same concerns to be able to reflect the contribution of the work more clearly. Major revision is done and it is highlighted in yellow. To simplify the contribution of the design a list is provided as below:

  • The system-level design of the proposed module is new in the field of communications and in compare with the similar works.
    1. It is covering low, mid, and high frequency bands with two separate antennas which are fairly practical in the application of the front-end Low Noise Amplifiers modules. In the applications such as LTE these bands are principal and the proposed module is a solution for these applications. That is the reason of such the specific configuration.
    2. It is offering two termination paths which are new in compare with the similar works fixing the safety which is discussed in the paper.
  • The proposed branched antenna is an approach to optimize the results for the application and it is compared with the conventional antenna.

Following the mentioned contributions of the work, the proposed module in this particular of the field of engineering is a new design systematically and practically. Following the reviewer’s concerns the manuscript is revised to reflect the steps that are taken for it. Also, the mentioned concerns about the sentence with “sizing” is concerned in the revision of the paper. An additional analysis is proposed for Fig. 4 as well.

Reviewer 2 Report

This papper reports on a 0.617-2.7 GHz Highly-Linear High-Power Dual Port 15 2 Throws Antenna Switch Module (DP15T-ASM) with Branched-3 Antenna Technique and Termination-Mode.

My comments and suggestions are as follows:

  1. The English needs intensive revision. I have marked many places where either grammar errors or usage errors. Please be noted, these highlighted places do not cover all the errors in English. 
  2. Regarding the technical part. (1) These are very few literature review on this topic. The authors only mentioned very limited techniques or work in the literature. It seems not sufficient to cover the-art-of-the-state of the progress. (2) what kind of simulation tools are used for the performance characterization? (3) the presentation of Figure 6 and Figure 9 is poor and must be improved. (4) The comparison of power handling capability in Table 1 is missing. (5) what is the relationship of Equations (1)-(5) to the work of this paper, and what is the logic relationship between these equations? (6) The process of Figure 3 shall be explained using quantitative description.

Author Response

Concern A) The English needs intensive revision. I have marked many places where either grammar errors or usage errors. Please be noted, these highlighted places do not cover all the errors in English

Author response:  The reviewer is right and the English of the paper is revised.

Author action: We updated the manuscript by performing a grammar check and revise the Grammar errors.

Regarding the technical part (B):

Concern 1) These are very few literature review on this topic. The authors only mentioned very limited techniques or work in the literature. It seems not sufficient to cover the-art-of-the-state of the progress.

Author response:  The reviewer is right and more techniques and works in the literature are added to the paper.

Author action: We updated the manuscript by adding additional references and contents to the introduction in the lines from 37 to 49.

Concern 2) what kind of simulation tools are used for the performance characterization?

Author response:  Cadence Virtuoso and ADS tools are used for the design. For the measurement, several devices are used such as RF signal generator, Network analyzer, Oscilloscope, and power supply. Also, the SP files are extracted using the network analyzer and it is plotted with a viewer software.

Concern 3) the presentation of Figure 6 and Figure 9 is poor and must be improved.

Author action: We updated the manuscript by improving the photos. In the revision their numbers are changed to Figure 8 and Figure 10, respectively. Also, descriptions related to the figures are improved in the manuscript.

Concern 4) The comparison of power handling capability in Table 1 is missing.

Author action: The comparison of the power handling is also added to Table 1.

Concern 5) what is the relationship of Equations (1)-(5) to the work of this paper, and what is the logic relationship between these equations?

Author response:  Equation (1) is mentioning the reason of the number of the stacks which is due to the power handling. Equation (2)-(5) are mentioning the analysis, which resulted in the proposed branched antenna. Equations (3)-(5) are mentioned to result in the final equation which is (5) to show that the variation of the H results in variation of the L which is the inductance effect. By a high variation for the inductance, RF conjugate matching varies as well which results in a an inconsistence operation for every throw.

Concern 6) The process of Figure 3 shall be explained using quantitative description.

Author action: We updated the manuscript by revising the paper and adding quantitive descriptions in the lines 254 to 263.

Round 2

Reviewer 1 Report

No comments. A good efford has been made

Author Response

Dear Reviewer,

Thank you very much for you comments during the submission process.

Best Regards,

Kang-Yoon Lee

Reviewer 2 Report

Thanks to the authors for the resvisions.

There are a few places where English may need to be corrected. For instance, in the abstract part: "including low- frequency bands of 617-960 13 MHz, mid-frequency bands of 1.4-2.2 GHz, and high-frequency bands of 2.3-2.7 GHz." frequency bands here may be changed to frequency band, since only one band is mentioned for each range.

Author Response

Dear Reviewer,

The mentioned English errors are corrected in the abstract of the paper, following your comment.

Thank you very much for your comments during the submission process.

Best Regards,

Kang-Yoon Lee